# Guidelines for Renewal and Securitization of a Critical Infrastructure Based on IoT Networks

Cristina Villar Miguelez [1], Victor Monzon Baeza [2,*], Raúl Parada [3] and Carlos Monzo [1]

1 Faculty of Computer Science, Multimedia and Telecommunications, Universitat Oberta de Catalunya (UOC), Rambla del Poblenou 156, 08018 Barcelona, Spain
2 Interdisciplinary Centre for Security, Reliability and Trust (SnT), University of Luxembourg, L-4366 Esch-sur-Alzette, Luxembourg
3 Centre Tecnològic de Telecomunicacions de Catalunya (CTTC), Av. Carl Friedrich Gauss, 7-Edifici B4, 08860 Castelldefels, Spain
* Correspondence: victor.monzon@uni.lu

**Abstract:** Global warming has increased uncertainty regarding managing traditional water supply systems. Unfortunately, there is a need for the smart management of water supply systems. This work aims to design a solution for renewing and securing critical infrastructure that supplies water and provides water purification inside the range of applications of Industry 4.0 for Smart Cities. Therefore, we analyze the renewal requirements and the applicable use cases and propose a solution based on IoT networks for critical infrastructure in the urban environment. We describe the architecture of the IoT network and the specific hardware for securing a water supply and wastewater treatment chain. In addition, the water level control process for the supply chain and the system that ensures the optimal level of chemicals for wastewater treatment are detailed. Finally, we present the guidelines for infrastructure operators to carry out this operation within Industry 4.0, constituting a development framework for future research on the design of Smart Cities.

**Keywords:** Industry 4.0; critical infrastructure; water management; IoT network

## 1. Introduction

Society is in a connected world in which technology and digitization are increasingly important. There is a more significant depletion of natural resources as the population increases. Quality and renewed water supply chains are essential for reducing drinking water consumption. Furthermore, global warming would increase uncertainty regarding managing traditional water supply systems according to [1]. Hence, water systems become critical infrastructure for a city, and it is essential to renew and make them compatible with future technologies such as artificial intelligence, as proposed in [1]. This helps to prevent possible leaks or failures of infrastructures. Digitally transforming the traditional city and making it a Smart City translates into making multiple advances in all internal processes.

Humans are causing a high negative impact on the environment with actions such as increased water consumption [2] and wastewater discharged into the sea without proper treatment [3]. Therefore, renewing the water supply chain and cities' wastewater treatment is necessary to help reduce the negative environmental impact. For this, we will apply the advantages of Industry 4.0. This is part of the general Smart City trend, in which automation and real-time monitoring are the protagonists. Specifically, in this study on the application of Industry 4.0 in the water and wastewater sector [4], it is proposed to integrate different systems to take advantage of data synchronization from various sources to reduce water consumption.

To fully apply digitization in the Industry, it is necessary to renew the environments related to scathing criticisms that provide society with an essential good, such as the case of water supply and wastewater control [5–7]. These infrastructures must ensure

the total availability of the service and very low latency. The article [8] explains NB-IoT technology's benefits for critical infrastructures. As cybersecurity plays a vital role in critical infrastructure, it is crucial to consider the security guidelines recommended in the study [9]. The study in [10] proposes to organize the different components of critical infrastructure networks on independent islands with high availability in such a way that a drop in one of these islands does not affect the rest.

With the advancement of Industry 4.0, processes increasingly essential for the population are being digitized and renewed. The water level control in a dam in India is described in the following paper [11]. It explains how the water level is measured using IoT technology, which sends the data to a central control platform. The use of IoT technology helps to renew the critical infrastructures of a Smart City. Kodali, Rajanarayanan and Yadavilli [12] show examples of an IoT project for critical infrastructure, which proposes control of chemical parameters in the water and sends an alert if these parameters are not within limits previously established by the specialized worker. Moreover, research from Chellaswamy et al. [13] shows actions identified to carry out a cloud platform that receives monitoring data of wastewater. It is an Indian experiment using Arduino One hardware. This is useful for developing applications for Smart Cities. Finally, the article in [14] experiments with the real-time response of IoT devices using the NB-IoT protocol, which would improve the supply chain control process.

### 1.1. Background

Nowadays, the Internet of Things (IoT) is a key technology for modern society. All communication systems are being renewed to include IoT to improve communication. For example, in [15], IoT is included in maritime communications. The literature shown in this section is focused on different areas supporting our research, such as the Industry 4.0 concept, IoT application in Industry 4.0 and the water sector. Industry 4.0 is a concept where the IoT shall be the leading technology to develop products with their own intelligence acquired during manufacturing. This new approach for industry includes intelligent transport and logistics (smart mobility and smart logistics), which contribute to energy efficiency to connect or create a Smart City. Lom et al. in [16] show the keys to integrating Industry 4.0 as a part of a Smart City.

Alcácer, V. et al. in [17] review the key within Industry 4.0 to lead the disruptive impact on manufacturing companies which will allow the smart manufacturing ecosystem paradigm. The enabling technologies and systems in the manufacturing environment are described. One of these technologies is cloud computing in conjunction with IoT, as proposed in [18]. Hence, recent studies report IoT-based productivity gains in manufacturing plants using sensors for data acquisition and supervisory control. However, bidirectional communication between these sensors is daunting because of the restrictions imposed by the variety and incompatibility of their interfaces and communication protocols. This problem increases the price of these solutions.

Within the group of IoT technologies, in this work, we have selected NB-IoT and the lower-power WAN (LPWAN) family of technologies and protocols. A comparative review of these protocols and their performance can be found in [19]. Leslie et al. argued for the potential of linking these technologies in [20]. The decision factors between technologies can be found in [21]. In this context, for Smart Cities, we have selected several works that help to contextualize the need for renovation already carried out in other city sectors. For the case of the agricultural sector A in [22], they present the trends and needs for its renovation towards Smart environments, while Monzon et. al in [23] present a satellite-based architecture with IoT to transport data from remote sites to the city. Another application for the renovation of the agricultural sector can be found in [24]. As for the energy sector, we have solutions in [25–27]. Technical guidelines have already been proposed for building a secure NB-IoT environment in [28]. However, they do not consider the regulatory and standardization topics. Perez et. al in [29] propose WPAN and LPWAN hybrid networks used for global

emergencies in Smart Cities. In the case of Smart water, [30] compares state-of-the-art methods to predict the water level in Catalan reservoirs.

To deal with the renewal of the infrastructures for industry, it is important to define the link between industrial designs and the effect that this renewal has on the sustainability that marks a smart city. Kuys, Koch and Renda in [31] perform a study of this performance for industrial design, conducted by 190 respondents from 53 countries, to establish the present state of industrial design practice globally and to better understand the priority of sustainability. The study performed by Cotrino, Sebastian and Gonzalez-Gaya in [32] uses a systematic literature review approach to determine the current research progress and future research potential of Industry 4.0 technologies to achieve manufacturing sustainability.

Within the industry, we find the water sector to be one of the most valuable because of the natural resource used for everyday living. Alabi, in [33], extends the concept of Industry 4.0 to Water 4.0 to include the water industry. In [33], an integrated business model for Water 4.0 is proposed. Different cloud platforms controlled through IoT are proposed to measure water quality in [34,35].

Jeurkar et al. in [36] focus on monitoring water use in a flat system. Other solutions, such as those shown in [37–39], are proposed to make the water manufacturing process smarter. However, these propose to require environmental awareness to distribute the sensors and high data processing from the sensors, which are not feasible for critical infrastructures.

For the development of the digitization of the industry, it is necessary to set some guidelines. There are many digitization solutions, so it is necessary to frame design guidelines at a high level. This article is to standardize the design criteria of a critical infrastructure environment. For this reason, a deep study of current ISO standards has been conducted.

### 1.2. Our Contribution

Before the previous background, this article proposes an architecture and the type of communications protocol selected to renew the infrastructure of a water supply chain and wastewater treatment, considered critical infrastructure. For this purpose, we include the following:

- Study, analysis and definition of requirements for critical infrastructures based on IoT environment.
- Solution for the renewal and securitization of the water supply and purification system based on IoT networks: it includes details of the architecture and the integration model of the techniques that are part of the monitoring and control solution set, as well the industrial supply chain and wastewater treatment.
- An intelligent water management framework for Smart Cities. We define the architecture, the bases and the necessary regulations to carry out technical projects.

The European directive 2008/114/CE on identifying and designation critical European infrastructures and evaluating the need to improve their protection is in force at the European Union level. This directive has been used to dictate the guidelines used in critical infrastructure such as the one proposed.

At the standards level, in 2018, the specific International Organization for Standardization (ISO) created indicators of city services and quality of life in its ISO 37120 [40]. In 2019, the ISO 37122 [41] incorporated two even more specific standards for Smart Cities based on the parent standard ISO 37120, which was on indicators for a Smart City. ISO 37123 [42] indicated resilient cities capable of maintaining their regular operation despite unknown disturbances. In a Smart City, certain specific operating parameters should be able to be measured.

At the level of cybersecurity regulations, we have UNE-EN ISO/IEC 27001: 2017 [43], which deals with the requirements requested in RDL 8/2011 on protecting critical infrastructures. Specifically, these common objectives are to protect: data confidentiality, data integrity and information availability.

The article is organized as follows. An introduction of this paper is made in Section 1. The requirements to define a critical infrastructure are shown in Section 2. The proposed IoT-based architecture is described in Section 3. In Section 4, KPIs to measure the proposal are shown, while Section 5 shows use cases. Finally, we conclude with some remarks in Section 6.

## 2. Requirements for Critical Infrastructure

All critical infrastructures, especially those related to the water supply chain, must meet, on the one hand, a series of technical or technological requirements to be renewed and, on the other hand, the criteria or conditions set in specifications and standards according to the regulation.

The requirements related directly to infrastructure are as follows:

- **High availability**: The availability of systems related to water supply and treatment must be at least 99%, which means that business activity should continue beyond 99% of the total activity time. Systems update: systems must be updated without affecting business availability. Upgrades before completion will have been tested on preproduction equipment to ensure a high probability of success in the production environment on the deployment day.
- **Grouping and analysis of the data obtained by the sensors**: A platform must be created to concentrate the data from all the sensors and homogenize them in the same format. Once the data have been grouped and standardized, they can be processed by the following system: the computer system, which will execute actions based on specifically configured alarms.
- **Daily maintenance of the solution in 24 × 7 mode**: A system maintenance plan should be included by operators with remote supervision and local on-site actions in case of high-level incidences. Operators should go to the water supply and water purification plant based on the detection of the incident that they have investigated with the remote monitoring platform.

On the other hand, security requirements for the proposed architecture are extracted from Standard ISO 27002 [44].

- **Documentation of operating procedures**: Any process related to the maintenance of the solution and operation, whether due to a configuration change or a task to update the firmware of equipment, must have its procedures documented. Specific plans must be established where the actions to be carried out and the authorizations to request the difference in the applicable network are planned.
- **Separation of environments:** This means having different and segmented infrastructures, whether physical or virtual, to deploy development environments, to develop new applications, to deploy test environments for using them, for example, to carry out bug update tests, and finally, there is the production environment that is in charge of running the product business. Any failure in the production environment supposes a stop in some of the services provided by the supply chain and water treatment.
- **Continuity of the security and infrastructure solution:** There will be an infrastructure with high availability with several firewalls in cluster mode, in such a way as to ensure the redundancy of the systems that support the solution for critical infrastructure. The details of the infrastructure to be deployed are detailed in Section 3 of this paper. In addition, network security policies for access control are included. Thus, segmentation mechanisms must be implemented for the different networks, avoiding traditional flat networks where users could access any infrastructure service.
- **Plans for monitoring SLAs**: The water supply and purification solution provider must comply with the service-level agreements (SLAs) according to the business parameters, and in case of noncompliance, the appropriate penalties will be shown. The transfer of information from a previous provider to a new provider would be done through secure protocols and encrypted data before the signing of confidentiality

contracts. The service provider must be requested to comply with the certification based on the ISO 27001 standard before any incident.

## 3. Computing Architecture

The architecture proposed for the solution comprises two subsystems: a computing core group and an action and measuring group.

### 3.1. Computing Core Group

The computing core group is the main architecture of the solution. This part of the proposed system will collect all the data from the IoT sensors and actuators of the architecture, named in this paper as the action and measuring group. The computing core group will receive data through 4G radio technology and store it in DDBB deployed in the data center servers. The extracted information will be studied through the data analytics processors, and the configurations at the software level of our system can be applied. the computing core group is based on a set of servers, switches, routers and firewalls to comply with current security standards for the industrial sector. This subsystem, as shown in Figure 1, is made up of the following platforms:

- **Centralized management IoT platform**: It is the central node for analyzing the data collected by the sensors and allows controlling the actuators' actions.
- **Monitoring, control and dashboard platform**: Policies are configured at a central level and fed by the data the IoT platform has analyzed. It can be accessed remotely through a secure connection. This platform will allow the visualization of the data in a control panel that will be accessed via the web.

  Moreover, the *computing core group* will be developed in different IT environments:

- **Production environment**: It is the one in charge of the execution of the tasks of the solution, such as the control of the water level in the supply chain or the management of the level of chemicals for the sanitation of wastewater.
- **Test environment** (also called preproduction environment): It is the environment before the production environment in which the tests of new software updates of the critical infrastructure are carried out. Any changes to the software are tested in this environment before running in the production environment.
- **Development environment**: It is one in which pilots are run for new functionalities of the solution covered by R&D studies.

The production and preproduction environments will have similar computing capabilities, since the preproduction environment detects and prevents possible failures before deploying in the production environment by avoiding incidents and prolonged service downtime. The system operator could deploy both environments on private cloud provider infrastructure.

The development environment will be deployed in a public cloud provider and accessible through private networks; it will not be given access through a public Internet address. For this environment, public cloud infrastructure has been selected to lower the cost of the solution and speed up the time of creation, testing and configuration of pilots for R&D studies. Suppose these pilots were optimal and passed all the latency and resilience tests. In that case, they could be applied to the preproduction environment and later production, improving the general application of central computing. These drivers could range from enhancements to increasing the artificial intelligence of data analysis to improvements to the data visualization environment for the dashboard.

*The computing core group* architecture is shown in Figure 1. Next, this subsystem is described, starting from the operator's private network necessary for the remote connection of the operators and continuing with the layers of the solution infrastructure, which are the routing layer, security layer, LAN switching layer and processing layer for data control and analysis.

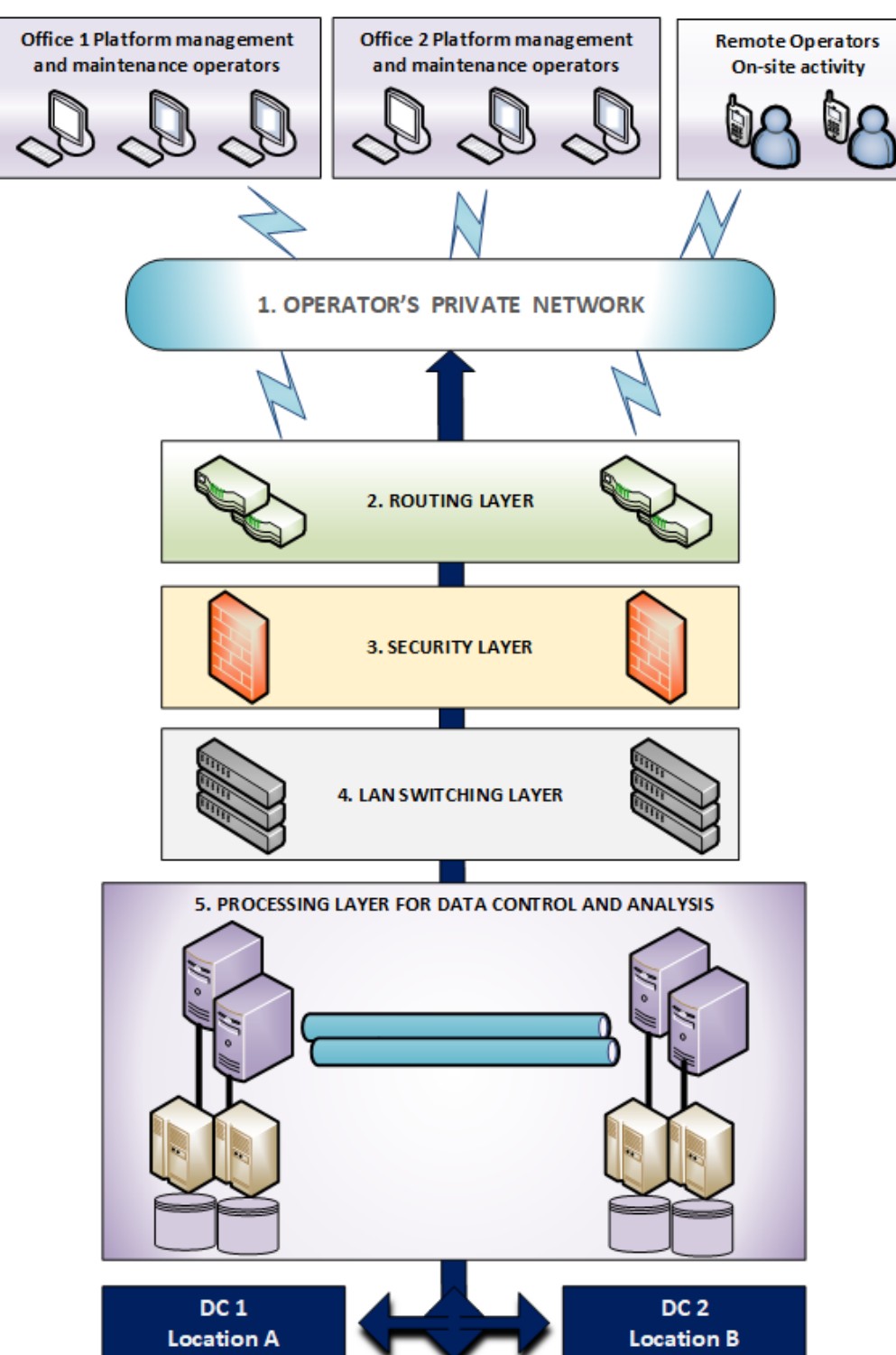

**Figure 1.** Computing core group architecture.

**Core group—operator's private network:**

For the remote connection of the operators to the central computing group and the connection between the data collection and action group with the main computing group. In this solution, we need low-consumption sensor networks with a wide coverage range that are robust against interference. In addition, it implements a strong security protocol with low latency between the IoT devices and the central management platform. The cost of the sensors must be as low as possible to facilitate the investments for the adoption. For this

reason, the choice was made within the specific solutions within the LPWAN category. We specifically opted for those that reuse the existing cellular network structure to avoid investments in additional antennas. On the other hand, the most appropriate technologies for the design under the study of the water supply and purification of a Smart City will be those that use licensed spectrum to avoid interference. With the previous descriptions, it only remains to select the protocol between the NB-IoT or LTE-M options. Both use the security standards of the mobile network, essential for use in critical infrastructures. However, using NB-IoT was decided to have a lower cost in acquiring the devices. In addition, studies were found that use NB-IoT devices to monitor critical infrastructures, such as [14], in which actuators are monitored and controlled with a latency time of 8 seconds in indoor scenarios and 2 seconds in outdoor settings. For this reason, NB-IoT was also selected to implement the proposed solution.

A previous study in [45] has considered protocols and technologies for IoT. We collected the data concerning critical infrastructures in Table 1, from which the architecture based on NB-IoT was selected over other technologies in the IoT framework, based on priority on the security required by critical infrastructures. NB-IoT is not as sensitive to interference as LoRaWAN, and its encryption protocol is AES-256 versus LoRaWAN's AES-128 [45,46]. LPWANs can be vulnerable to security attacks, but NB-IoT offers sufficient security for certain applications if implemented with strong security policies and enforcement. Therefore, the lower interference level of NB-IoT due to the use of licensed radio frequency bands makes the proposed infrastructure more robust than LoRaWAN. The LoRaWAN and Sigfox options were discarded because they work in a spectrum without a license. Therefore, any user could use this spectrum, causing interference that would not allow a high level of availability of the intended service. Some simulations on the deployment and use of NB-IoT technology are summarized in [47], indicating latency in data transmission depending on whether they are indoors or outdoors. An NB-IoT traffic simulator in Smart City environments shows an average transmission delay value of 34.781 ms and 6.929 ms for indoor and outdoor environments, respectively. For average total delay, the values for Smart Cities shown in [47] are 239.876 ms for indoor and 85.676 ms for outdoor. These values are collected in Table 2. We can use them as a reference for latency in future deployments of the proposed architecture in this paper.

**Table 1.** Comparison of LPWAN technologies (Based on the study in [45]).

| Parameter | NB-IoT | LoRa | SigFox | LTE-M |
|---|---|---|---|---|
| Standard | 3GPP Rel. 13,14 | LoRa Alliance | ETSI LTN | 3GPP Release 14 |
| Frequency | 900 MHz | 868 MHz | 868 MHz | Cellular Band |
| Bandwidth | 200 KHz | 250 KHz | 100 Hz | 1.4–20 MHz |
| Security | NSA AES 256 | AES 128b | Optional Encryption | AES 256 |
| Topology | Star | Star of stars | Star | Star |
| Throughput (max data rate) | 200 kbps | 50 kbps | 600 bps | 4 Mbps |
| Range | 1–5 km | 2–5 km | 3–10 km | 1–5 km |
| Power Consumption | Medium–Low | Very Low | Low | Medium |
| Battery | 10 years | 10 years | 12 years | 2 years |
| Deployment Cost | Moderate | Moderate | Moderate | High |

**Table 2.** Latency value for Critical Infrastructure in a Smart City as a reference.

|  | Average Transmission Delay (ms) | Average Total Delay (ms) |
|---|---|---|
| Indoor NB-IoT | 34.781 | 239.876 |
| Outdoor NB-IoT | 6.929 | 85.676 |

On the other hand, according to [48], the latest 2021 statistics indicate the higher adoption of NB-IoT over LPWAN technologies for deploying critical infrastructure, as proposed here. For all these reasons presented so far, NB-IoT is chosen over other alternatives in IoT technologies.

The selected private network operator must ensure that 4G coverage is necessary to deploy the IoT solution based on the NB-IoT protocol.

**Computing core group—routing layer:**

It is made up of four router devices that allow communication at the IP level with the private network of the selected operator. These routers have activated IP filtering lists, allowing the solution's range of IPs to pass through, improving the solution's security level by preventing unauthorized IPs from passing through. Dedicated FTTH connects the routers to the operator's network with a fully diversified primary, and a backup connection with fiber runs at different locations to guarantee the operation of the solution in the event of a fall on one of the two roads.

**Computing core group—security layer:**

This subsystem will comprise a cluster of two firewall devices in an active–active high availability configuration, segmenting the network into different virtual routing and forwarding (VRF) traffic between said networks. VRF between the management and operation traffic of the solution, the sensorization data traffic and the actuator traffic will make the differentiation.

**Computing core group—LAN switching layer**:

Switches will be used to distribute traffic at Layer 2 to bring it to the processing layer for control and analysis.

**Computing core group—processing layer for data control and analysis** :

This element includes the "IoT platform" and "Monitoring, Control and Dashboard platform".

The requirements for the "IoT platforms" of the proposed architecture are divided into two large groups according to the functionalities that they will contribute:

- IoT platform for water supply control: Designed to control the sensors and actuators of the water supply system. They are all the policies that have been developed to analyze the data provided by the water supply sensors, such as the water level sensor. This platform also includes the actions programmed to execute the orders to the actuators based on guidelines drawn up by the operators in charge of maintenance and management of the infrastructure.
- IoT platform for water purification system: Designed to control the sensors and actuators dedicated to the water purification chain. It allows the creation of action policies on the actuators of the solution water purification chain.

Regarding the "Monitoring, Control and Dashboard platform", it will have the following functionalities:

- Visualization of the data of the different KPIs in web mode is designed for this solution. KPIs are detailed in Section 4 of this paper.
- Analysis of the data received from the sensors.
- Creation of reports to monitor the operation and possible incidents.

The *processing layer for data control and analysis* will be formed at a physical level by the following:

- Servers: Those in charge of displaying information from both the IoT platforms and the control platform and dashboard. These front servers will be a total of two clusters

in high availability. These servers will be contracted in virtualized mode, which will help the flexibility of the growth of the solution according to the needs of the business.
- Databases (DDBB) will be the store. They can be relational SQL databases or nonrelational databases such as MongoDB. The databases will also be two clusters in high availability and connected at level 2 to ensure their synchronism.
- Data copy system: A copy of the database data will be available in order to have them in case of incidence or corruption of the database data in production.

An example of commercially deploying the "Central Control and Analysis platform" is the FIWARE solution created by the European Program for the Development of the Internet of the Future (FI-PPP).

We recommend for the processing layer the use of data type XML. XML data support the exchange of information between computer systems and are easy to process. We also propose to use MongoDB as DDBB because NoSQL databases offer higher performance. In [49], a solution for data models for Internet applications is described with these XML data and MongoDB.

### 3.2. Action and Measuring Group

Sensors and actuators in this group communicate with several gateway devices, which talk to the *computing core group* of the solution. We can see the *action and measuring group* architecture in Figure 2.

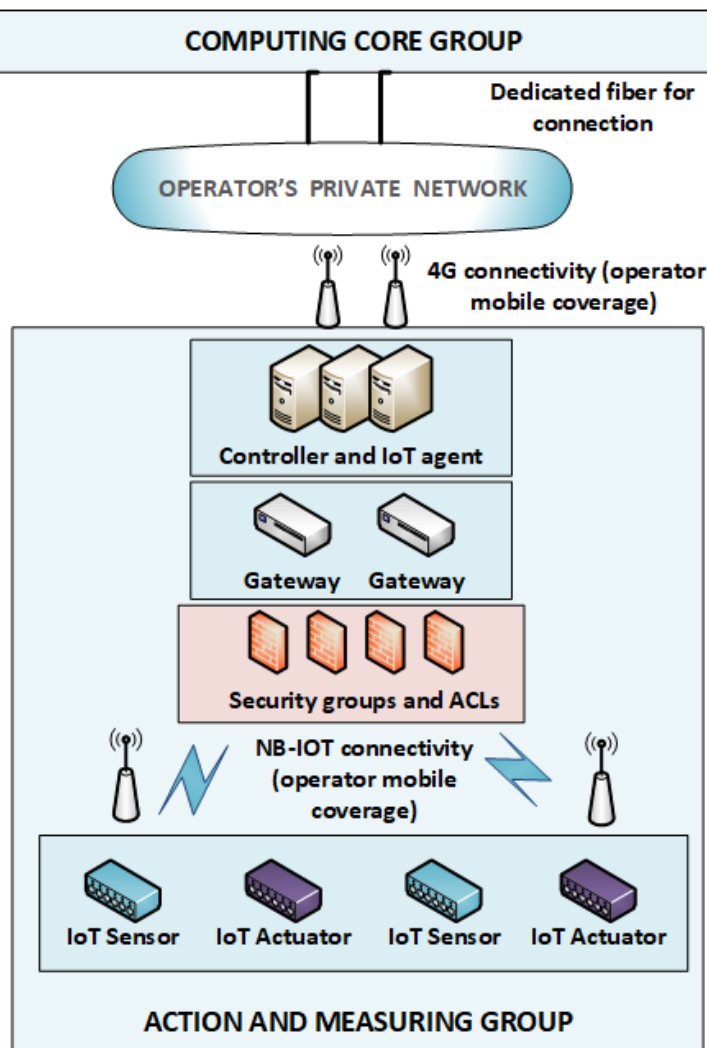

**Figure 2.** Action and measuring group architecture.

It comprises the levels of sensorization and action and is responsible for measuring any event that happens in the physical environment. It is the layer close to the physical elements of the supply chain and water treatment. More details about this system are as follows: action and measuring group

***Sensorization level:*** It is the level related to the physical layer in which the NB-IoT devices of our solution are located. Some examples of NB-IoT sensors that could be used in our solution can be those from the manufacturer Sosteco, Movisat, or Kunak. In addition to the sensors, gateways communicate between the IoT network and the central computing group. IoT controllers would also be included for orderly data transmission to the central computing group.

***Action level:*** It is the level related to the physical layer in which the NB-IoT devices of our solution are located. Unlike the class of sensorization, they are the devices in charge of making actions based on the configuration indicated in the central computing node. Some examples of actuators are door-opening regulation devices in the water supply chain, a device for controlling the number of chemicals to be included in the wastewater for proper treatment, or even devices that measure the noise level they make.

Elements of the group in Figure 2 are as follows:

***Physical devices:*** Actuators and IoT sensors. They will connect wirelessly with the gateway and through the security groups and access control lists (ACL) configured on the network. Manufacturers of NB-IoT hardware devices are classified according to the type of market they are targeting; there are those specialized in Smart-Cities-related parameters such as water quality control, for example, Libelium and Kunak, or those specialized in electricity consumption control, such as Schneider. A price reference for an NB-IoT water chemical sensor device is about EUR 50 per unit for the Kunak K101 model [50].

***Gateway devices:*** They communicate with the controller and IoT agent, and through them, the information is sent to the operator's private network through 4G connectivity. The availability of 4G coverage in the area must be ensured before proceeding with its deployment.

## 4. Key Performance Indicators (KPI)

To assess and accept the architecture of Smart Water infrastructure such as the one proposed in this work as an example of the Internet of Time-Critical Things, we use KPIs (key performance indicators) defined by a series of standards and norms in Section 1. The KPIs selected are summarized in Table 3.

**Table 3.** KPI.

| | KPI | Description |
|---|---|---|
| PED | Protection of environmental diversity | To control that the water's purification is high enough to avoid altering the ecological environment. |
| PP | Pollution protection | To control the level of pollution generated by the water supply and purification solution. |
| CEE | Control of energy efficiency | To measure the amount of energy consumed by the population reached by the water supply solution. |
| SWL | Savings in water leaks | To measure the leaks that may exist in our supply chain. |
| ACP | Amount of water consumed by the population | To measure the amount of water consumed by the people (by a total number of inhabitants) based on the amount of water supplied by our solution. |

**Table 3.** *Cont.*

| | KPI | Description |
|---|---|---|
| PR | Population reached | To measure the population reached by the wastewater collection facility. |
| LPT | Level of water purification treatment | To measure the level of water purification treatment according to the specialization of the treatment. |
| AIS | Average interruption of service | To measure the average annual hours of interruption of the water supply service. |

These KPIs measure the protection of environmental diversity, the level of pollution generated by the water supply and purification solution, energy efficiency, the savings in water leaks in the supply chain and the amount of water consumed by the population, as well as the level of water purification treatment and the population reached by the wastewater collection facility. Finally, they measure the average annual hours of interruption of service.

## 5. Use Cases Results

Specific use cases using the aforementioned proposed architecture are described as an example below.

### 5.1. Water Level Management

The use case for water level management for the proposed architecture is shown in Figure 3. Sensors and actuators are located in the water supply pipe every 100 meters. The level sensor sends the data through the NB-IoT protocol to the central IoT platform. Whether the water level is at optimal levels or not, a signal will be sent to the actuator to open or close the pipeline gate. In the example of Figure 3, it can be seen that sensor 2 detects that the water level is much higher than sensor 1, which is why a signal is sent to actuator 2 to close the water damper and regulate the water flow. This allows for equalizing the water level at the different gates of the water supply chain.

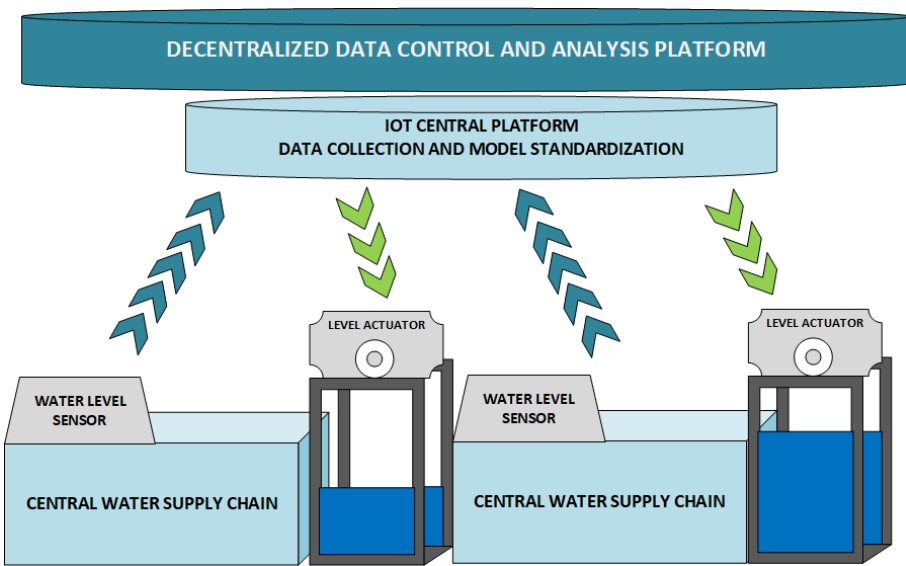

**Figure 3.** Water level management procedure.

### 5.2. Chemical Level Management in Water

According to the proposed solution, we can see in Figure 4 the use case for chemical level management in water.

This is an example of the sensor and actuator solution for wastewater treatment. Sensor number 1 of Figure 4 shows how it detects more microorganisms than desired in the wastewater. For this reason, from the central IoT platform, the flow of action data are sent so that the chemical that controls the number of organisms is introduced into the water. This process will be repeated many times until the level of microorganisms in the water is the desired one. Before repeating the process, it is necessary to wait a minimum time for the chemical to take effect and dilute in the water.

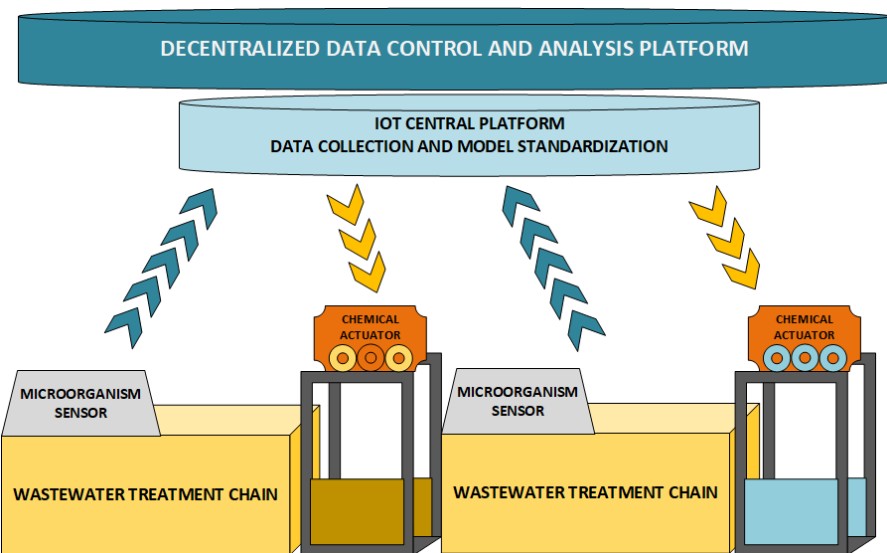

**Figure 4.** Chemical level management procedure.

The advantages obtained with the proposed infrastructure are shown next:

- Low power consumption: Using the NB-IoT communication protocol consumes little power for the IoT sensors.
- Wide range of sensors: Using an LPWAN solution achieves long ranges of communication with the sensors of the solution through the cellular network.
- Low cost in economic investment by reusing the infrastructure of existing 4G radio stations in the network of the selected operator.
- Lower cost due to NB-IoT sensors: Cost per NB-IoT sensor device is lower than Sigfox, LTE-M, or LoRa devices.
- Open-source flexibility of the management platform: Using an open-source platform improves the availability of applications for analyzing data received from IoT sensors.

The drawbacks are described next:

- Sparse standards: Further advances in standardization are needed to allow greater IoT network solutions.
- IoT sensors and actuators: There are few certified sensors on the market due to the lack of completeness of the standards, which means an increase in the price of all LPWAN devices.

## 6. Discussion

This article provides a common framework for designing a critical infrastructure that will comply with the standards established in Industry 4.0. It is intended to create a technical design guide to digitalize productive environments in the industrial sector. It is not about creating a unique and inflexible design but about laying the foundations for an Industry 4.0 environment. Below, we recommend guidelines to digitalize and renew a critical infrastructure within the industrial environment and some future improvements.

NB-IoT protocol and 4G mobile network coverage were chosen as the best technologies for the critical infrastructure renewal solution through IoT networks. NB-IoT uses

all the communication security protocols inherent in the mobile network, and the level of deployment of 4G coverage is high. The place where the solution is deployed will likely have sufficient 4G coverage. With the progress of standardization in 5G protocols, the natural evolution for IoT networks will be to use the advantages offered by the 5G network. Currently, the characteristics and needs of an IoT network are solved with the use of NB-IoT or LTE-M.

5G technology enables lower latency (uRLLC) and massive machine-type communications (mMTC), which would improve connectivity for the total number of IoT monitoring sensors required for critical infrastructure. However, as the number of sensor devices in our infrastructure grows, network design will need to be improved to meet the new lower latency requirements. This is where the use of 6G connectivity would come into play [51]. The improvements that 6G networks would bring and that would be applied to critical infrastructures that are usually located in areas with poor mobile coverage are, precisely, the improvement of coverage towards global coverage through the effective integration of satellite and submarine communication networks. Additionally, they provide a higher capacity than 5G networks, which would be improve by 1000 times through the use of Terahertz frequency and spatial multiplexing.

## 7. Conclusions and Future Lines

The use of hyperautomation and artificial intelligence systems is identified as another future implementation to be applied in our design to carry out preventive maintenance on old parts before the occurrence of a probable incident. The advantage of using hyperautomation is establishing robotic processes that control the water level of the supply network based on the data analyzed by artificial intelligence, which would largely avoid human action, and hence possible human failures. However, it would always be necessary to have technical operators available for potential unforeseen incidents, as it is a critical infrastructure in which service availability to citizens is the most important thing.

To propose future connectivity for the architecture model, we propose the 6G network, which will be able to meet future requirements, as more sensors will be needed to be connected simultaneously. However, it should be noted that there is no 6G standard, so we will have to wait for the technology to mature and become standardized before it can be applied in critical infrastructure. Future researchers might use the proposed framework to design interdisciplinary architectures within the Smart Cities paradigm. The proposed architecture model can be used to establish the minimum requirements for dedicated simulators to study the operation of IoT networks dedicated to water distribution systems, such as the EPANET simulation model proposed by the US government. This simulator is used to design new water supply infrastructures, to study water quality, establish the number of water pumps and even define the contamination models in the supply chain. Our architecture model could be used in future investigations with EPANET [52]. To provide an IoT monitor solution that meets the required specifications of the water supply chains.

**Author Contributions:** Conceptualization, C.V.M. and V.M.B.; investigation, C.V.M. and V.M.B.; writing—original draft preparation, C.V.M. and V.M.B.; writing—improving and correcting original version, C.V.M., V.M.B., R.P. and C.M.; writing—review and editing, C.V.M., V.M.B., R.P. and C.M.; supervision, V.M.B.; project administration, V.M.B. All authors have read and agreed to the published version of the manuscript.

**Funding:** This research received no external funding.

**Institutional Review Board Statement:** Not applicable.

**Informed Consent Statement:** Not applicable.

**Data Availability Statement:** Not applicable.

**Conflicts of Interest:** All authors declare that the research was conducted in the absence of any commercial or financial relationships that could be construed as a potential conflict of interest.

## Abbreviations

The following abbreviations are used in this manuscript:

| | |
|---|---|
| ACL | Access Control Lists |
| DDBB | Database |
| FI-PPP | Future Internet Public–Private Partnership |
| FTTH | Fiber To The Home |
| IoT | Internet of Things |
| ISO | International Organization for Standardization |
| IT | Information Technologies |
| KPI | Key Performance Indicator |
| LAN | Local Area Network |
| LoRa | Long Range |
| LoRaWAN | Long-Range Wide-Area Network |
| LPWAN | Low-Power Wide-Area Network |
| LTE | Long-Term Evolution |
| LTE-M | Long-Term Evolution Machine |
| mMTC | massive Machine-Type Communication |
| NB-IoT | NarrowBand IoT |
| SLA | Service-Level Agreement |
| SQL | Structured Query Language |
| uRLLC | Ultra-Reliable Low-Latency Communications |
| VRF | Virtual Routing and Forwarding |

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
