# Peer review of "Guidelines for Renewal and Securitization of a Critical Infrastructure Based on IoT Networks"

_smartcities, doi:10.3390/smartcities6020035_

Round 1

Reviewer 1 Report (Previous Reviewer 2)

Dear authors, please check the attached pdf file.

Author Response

Many thanks to the Reviewer for his/her exhaustive and detailed evaluation. We also appreciate that you have valued all the effort and work in the previous versions. We are happy that the work done has been appreciated in improving the initial version.

Reviewer 2 Report (New Reviewer)

In this paper, the authors first analyzed the renewal requirements and the applicable use cases of IoT networks on management of water supply systems. Then they proposed an IoT-based solution for critical infrastructure in urban areas. They also described the architecture of the IoT network and the specific hardware for securing a water supply and wastewater treatment chain. Additionally, they explained the water level control process and the system for optimal wastewater treatment chemical levels. Finally, the authors presented the guidelines for infrastructure operators to carry out this operation within Industry 4.0.

Strengths

1)      The authors introduced and analyzed the requirements for critical infrastructures based on IoT environment.

2)      The authors proposed an architecture for the renewal and securitization of the water supply and purification system based on IoT networks which includes the details of the architecture and the integration model of the techniques that are part of the monitoring and control solution set.

3)      The authors proposed an intelligent water management framework for Smart Cities, with definitions on the architecture, the bases and the necessary regulations to carry out technical projects.

Weaknesses

1)    The authors in the article used KPIs defined by a series of standards and norms to measure the performance of the proposed architecture, including the protection of environmental diversity, level of pollution and so on. However, the article only defines these standards without any certain rules and designs to efficiently use these indicators in real design. The authors also didn’t give out enough reasons about why these measurements should be used. Without these designs and rules, others might get quite confused about how to use these KPIs in practical applications, thus making the KPI system very much unlikely to be used in future designs.

2)    In the paper, despite the authors claimed that they proposed a solution for the renewal and securitization of the water supply and purification system based on IoT networks, their design is too general and lacks details. Although the authors claimed their design is only about creating a unique and inflexible design but about laying the foundations for an Industry 4.0 environment, certain details regarding specific design are still necessary. For example, in section 3.1 Computing core group, the authors only defined different parts and their intended functions, without their specific and detailed information, how they are designed, how their performances are measured, what makes the design greater than the other? All the composing parts such as routing layer, security layer and LAN switching layer have such problems, making the whole architecture very unpractical to follow.

3)    The authors fail to cite several past literatures highly related to this work (e.g., [1-3]) and clearly discuss the differences between them and this paper.

[1] Categorization and Criticality Assessment of Facilities of Critical Infrastructure. MLSD 2022.

[2] Cost-minimizing Mobile Access Point Deployment in Workflow-based Mobile Sensor Networks", ICNP 2014.

[3] Distributed information and control system for emergencies in critical infrastructures, AICT 2016.

4)    The article lacks certain experiments and has no comparisons with other state-of-the-art works, thus the proposed architecture lacks persuasion. For example, they should include baseline results and comparison performances of their own architecture under the same criteria. Only with such comparisons can one’s work be qualified for promotion and be set as foundation for future works. On the other hand, ablation study should also be included. With ablation study, we can be sure the different designs in the paper are effective, but not fancy and useless design. The authors should also include their novelty and creation in this particular work, why they are better or what’s new in their design that makes their proposed architecture a substitute for current design. These are all details the authors should include in their paper to make it more convincing.

Author Response

Many thanks for your precious time and efforts expended in reviewing our paper. We attempted to address all the remaining uncertainties. Our hope is that the paper was further clarified.

Round 2

Reviewer 2 Report (New Reviewer)

comments addressed. recommend acceptance. 

This manuscript is a resubmission of an earlier submission. The following is a list of the peer review reports and author responses from that submission.

Round 1

Reviewer 1 Report

The paper would have been better if there was at least 1 hypothesis. 

The paper is very theoretical in nature. Suggest to include a section on statistical analysis to justify the architecture.

Author Response

Many thanks for your precious time and efforts expended in reviewing our paper. We attempted to address all the remaining uncertainties. Our hope is that the paper was further clarified.

Please find attached our response to reviewer comments.

Reviewer 2 Report

Please see the attached PDF file.

Author Response

(The authors gave the same response as above.)

Reviewer 3 Report

The authors claim that the article provides a common framework for designing a critical infrastructure that will comply with the standards established in Industry 4.0. It is intended to create a technical design guide to digitalize productive environments in the industrial sector.

From Title of the paper, Abstract, all the way to Section 6: Discussion and Conclusions are not organized well. The authors should define the title clearly to reveal the purpose of this research. IoT networks are very important for smart cities. The authors should study latest technology to propose their framework for critical infrastructure evaluation as well as design of KPIs (Section 4).

For example, the authors should study 5G or 6G instead of 4G technology for IoT technology. It's the year 2023 now. 4G networks are out-of-date. The authors can use the following paper as reference:

https://doi.org/10.1016/j.icte.2022.06.006

Because of these issues and the lack of innovations to make this manuscript publishable, it should be rejected.

Author Response

(The authors gave the same response as above.)

Round 2

Reviewer 3 Report

1. The authors changed their paper's title, which is good to show this paper's property. But the submission system still shows the old title.

2. From the authors' first response to my first round of comments, the authors failed to address many issues, including technical spec, unknow country's regulations, unknow country's rules for a smart city, as well as missing the measure of level of infrastructure, etc. This reviewer does not think the authors have solid ground to ignore these issues in a research paper.

3. From the authors' second response to my first round of comments, the authors confirmed that the technology selection study had been done in their other work outside the scope of this paper. This manuscript is only limited to presenting the renewal guidelines. That is why this manuscript is fragmented, even in revision. 

4. The authors believed that their presentation was a logical order from the authors' third response to my first round of comments. This reviewer wants to suggest that the authors should check "Guidelines for Authors" from Smart Cities journal.

"The article should include the most recent and relevant references in the field. The structure should include an Abstract, Keywords, Introduction, Materials and Methods, Results, Discussion, and Conclusions (optional) sections, with a suggested minimum word count of 4000 words."

This reviewer does not see any research material or research methods as well as research results in this manuscript.

5. Additional comment to the authors' third response to my first round of comments. This reviewer wants to suggest the authors should check Smart Cities journal template, especially Abstract.

"A single paragraph of about 200 words maximum. For research articles, abstracts should give a pertinent overview of the work. We strongly encourage authors to use the following style of structured abstracts, but without headings: (1) Background: Place the question addressed in a broad context and highlight the purpose of the study; (2) Methods: briefly describe the main methods or treatments applied; (3) Results: summarize the article’s main findings; (4) Conclusions: indicate the main conclusions or interpretations. The abstract should be an objective representation of the article and it must not contain results that are not presented and substantiated in the main text and should not exaggerate the main conclusions."

This reviewer does not see (2)-(4), only (1) appeared in the original manuscript or in this revision.

6. This reviewer is very sorry that the authors' clarifications did not help to improve the quality and structure of this manuscript.

7. More suggestions to the authors as follows:

(1) Who is your target audience?

(2) Who will cite your paper in the future?

(3) What kinds of knowledge or information can be learned from your paper after spending time reading it?

Author Response

(The authors gave the same response as above.)

Round 3

Reviewer 3 Report

Thanks to the Authors for providing this third-round revision, which I do not expect to read this paper again and again. There still exist typos, format, inconsistency and grammar errors in the manuscript, such as Table 3 KPI and Reference list [40-52]. So let the post-production team of MDPI help you correct these problems for you.

Author Response

Many thanks for your precious time and efforts in this third round to improve our paper titled "Guidelines for Renewal and securitization of a critical infrastructure based on IoT networks". The new title to be considered is “A Smart Water Management framework based on IoT Networks”.

In this case, we have reviewed the minor typos comments from Reviewer 3, including the answer in the document “Response_Reviewer_3_Round3”. 
